# A Bayesian-Symbolic Approach to Learning and Reasoning for Intuitive Physics

## Abstract

Humans are capable of reasoning about physical phenomena by inferring laws of physics from a very limited set of observations. The inferred laws can potentially depend on unobserved properties, such as mass, texture, charge, etc. This sample-efficient physical reasoning is considered a core domain of human common-sense knowledge and hints at the existence of a physics engine in the head. In this paper, we propose a Bayesian symbolic framework for learning sample-efficient models of physical reasoning and prediction, which are of special interests in the field of intuitive physics. In our framework, the environment is represented by a top-down generative model with a collection of entities with some known and unknown properties as latent variables to capture uncertainty. The physics engine depends on physical laws which are modeled as interpretable symbolic expressions and are assumed to be functions of the latent properties of the entities interacting under simple Newtonian physics. As such, learning the laws is then reduced to symbolic regression and Bayesian inference methods are used to obtain the distribution of unobserved properties. These inference and regression steps are performed in an iterative manner following the expectation–maximization algorithm to infer the unknown properties and use them to learn the laws from a very small set of observations. We demonstrate that on three physics learning tasks that compared to the existing methods of learning physics, our proposed framework is more data-efficient, accurate and makes joint reasoning and learning possible.

## 1 Introduction

Imagine a ball rolling down a ramp. If asked to predict the trajectory of the ball, most of us will find it fairly easy to make a reasonable prediction. Not only that, simply by observing a single trajectory people can make reasonable guesses about the material and weight of the ball and the ramp. It is astonishing that while the exact answers to any of these prediction and reasoning tasks requires an in-depth knowledge of Newtonian mechanics and solving of some intricate equations, yet an average human can perform such tasks without any formal training in physics. Even from an early age, humans demonstrate an innate ability to quickly learn and discover the laws of physical interactions with very limited supervision. This allows them to efficiently reason and plan action about common-sense tasks even in absence of complete information (Spelke, 2000; Battaglia et al., 2013). Recent studies suggest that this ability of efficient physical reasoning with limited supervision is driven by a noisy model of the exact Newtonian dynamics, referred as the intuitive physics engine (IPE; Bates et al., 2015; Gerstenberg et al., 2015; Sanborn et al., 2013; Lake et al., 2017; Battaglia et al., 2013).

As sample-efficient physical reasoning is recognized as a core domain of human common-sense knowledge (Spelke & Kinzler, 2007); therefore an important problem in artificial intelligence is to develop agents that not only learn faster but also generalize beyond the training data. This has lead to a surge in works aimed at developing agents with an IPE or a model of the environment dynamics (Amos et al., 2018; Chang et al., 2016; Grzeszczuk & Animator, 1998; Fragkiadaki et al., 2015; Watters et al., 2017; Battaglia et al., 2016; Sanchez-Gonzalez et al., 2019; Ehrhardt et al., 2017; Kipf et al., 2018; Seo et al., 2019; Baradel et al., 2020). Among these, neural-network based learned models of physics (Breen et al., 2019; Battaglia et al., 2016; Sanchez-Gonzalez et al., 2019) tend to have good predictive accuracy but poor sample efficiency for learning. On the other hand, symbolic models (Ullman et al., 2018; Smith et al., 2019; Sanborn et al., 2013; Bramley et al., 2018) are sample efficient but fail to adapt or accommodate any deviation from their fixed physics engine.

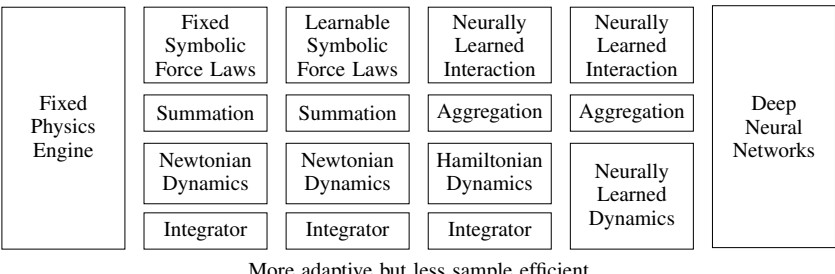

Figure 1: From left to right are rule-based to purely data-driven models of physics. Examples for each column are (1) (Smith et al., 2019), (2) (Ullman et al., 2018), (3) BSP (Ours), (4) OGN & HOGN (Sanchez-Gonzalez et al., 2019), (5) IN (Battaglia et al., 2016) and (6) (Breen et al., 2019).

Inspired by humans' highly data-efficient ability of learning and reasoning about their environment, we present Bayesian-symbolic physics (BSP), the first fully Bayesian approach to symbolic intuitive physics that, by combining symbolic learning of physical force laws and statistical learning of unobserved properties of objects, enjoys the sample efficiency of symbolic methods with the accuracy and generalization of data-driven learned approaches. In BSP, we pose the evolution of the environment dynamics over time as a generative program of its objects interacting under Newtonian mechanics using forces, as shown in figure 2. Being a fully Bayesian model, we treat objects and their properties such as mass, charge, etc. as random variables. As force laws are simply functions of these properties under the Newtonian assumption, in BSP we replace data-hungry neural networks (NN) with symbolic regression (SR) to learn explicit force laws (in symbolic form) and then evolve them deterministically using equations of motion. But a naive SR implementation is not enough: a vanilla grammar that does not constrain the search space of the force-laws can potentially have far worse sample efficiency and accuracy than a neural network. Therefore, we also introduce a *grammar of Newtonian physics* that leverages *dimensional analysis* to induce a physical unit system over the search space and then imposes physics-based constraints on the production rules, which help prune away any physically meaningless laws, thus drastically speeding up SR.

Our main contributions are threefold:

- We introduce a fully differentiable, top-down, Bayesian model for physical dynamics and an expectation-maximization (EM) based algorithm, which combines Markov chain Monte Carlo (MCMC) and SR, for maximum likelihood fitting of the model.
- We introduce a *grammar of Newtonian physics* that appropriately constrains SR to allow data-efficient physics learning.
- Through empirical evaluations, we demonstrate that the BSP approach reaches human-like sample efficiency, often just requiring 1 to 5 observations to learn the exact force laws – usually more than 10x fewer than that of the closest neural alternatives.

## 2 RELATED WORK

At a high level, the logic of physics engines can be decomposed into a dynamics module and a model of how the entities interact with each other depending on their mutual properties. These modules can be further divided into more components depending on how the module is realized. Using this break-down, we can categorize different models of physics based on what components of the model are learned. In figure 1, we compare some of the recent models of physics that are of closely related to our work. Starting on the right end, we have fully learned, deep neural-network approach used by Breen et al. (2019) that do not use any prior knowledge about physics and therefore learn to predict dynamics completely in purely data-driven way. In the middle are hybrid models that introduce some prior knowledge about physical interaction or dynamics in their deep network based prection model. These include interaction networks (INs; Battaglia et al., 2016), ODE graph networks (OGNs) and Hamiltonian ODE graph networks (HOGNs; Sanchez-Gonzalez et al., 2019). Since these approaches employ deep networks to learn, they tend to have very good predictive accuracy but extremely bad sample efficiency and therefore require orders of magnitude more data to train than humans (Ullman et al., 2018; Battaglia et al., 2016; Sanchez-Gonzalez et al., 2019). On

the other end of the spectrum (left) are the fully symbolic, rule-based physics models and engines (Smith et al., 2019; Allen et al., 2019; Wu et al., 2015; Ullman et al., 2018). While these methods are suitable for reasoning tasks, they lack the flexibility of the data-driven, learned models as they cannot generalize or adapt to any changes in the environment that their fixed physics engine simulates. For example, in such fixed models, inference can fail on physically implausible scenes and may require additional tricks to resolve such issues (Smith et al., 2019).

Symbolic regression has been used for general physics learning in many prior works ranging from Schmidt & Lipson (2009) that used SR to discover force laws from experimental data to the more recent work of Cranmer et al. (2020) on distilling symbolic forces from INs using genetic algorithms. Even more recently, Udrescu & Tegmark (2020) proposed an interesting framework AI Feynman, which recursively simplifies the SR problem using dimensional analysis and symmetries discovered by neural networks to discover the underlying physics equation that generated the data. The focus of these prior work has been to discover the underlying physical equations that directly leads to the observed data but unlike our approach, they do not target to allow for reasoning in physical environments, that is common task of interest in intuitive physics studies.

## 3 BAYESIAN-SYMBOLIC PHYSICS

Our framework, Bayesian-symbolic physics, combines symbolic learning of physical force laws with Bayesian statistical learning of object properties such as mass and charge. The environment is modelled by a probabilistic generative model governed by Newtonian dynamics. Physical laws are learnable symbolic expressions that determine the force exerted on each object, based on the position and properties of other objects. These properties might not be observed, so they are treated as latent variables and learned in a Bayesian fashion. The physical laws themselves have a prior distribution to organize the search space and discourage the model from learning physically meaningless laws; we call this distribution a *grammar of Newtonian physics*.

To learn with incomplete data, we are inspired by results from Ullman et al. (2018), humans are able to simultaneously predict the trajectory and update their inference about the properties of the object under new observation. This motivates an EM-based learning and inference method to fit BSP models. In the E-step, we obtain the distribution of entity properties by sampling from their posterior distribution using the current guess of force laws, and in the M-step, we use the samples from the E-step to perform SR to update the force functions. This enables BSP models to learn and reason in environment with incomplete information.

### 3.1 GENERATIVE MODEL OF THE ENVIRONMENT

We represent each entity $i \in \{1 \dots N\}$ by a vector of properties $z^i$, such as mass, charge, coefficient of friction, and shape, some of which may be unobserved. At each time step $t$, a state vector $\mathbf{s}_t^i = (\mathbf{p}_t^i, \mathbf{v}_t^i)$ is associated with each entity $i$, where $\mathbf{p}_t^i \in \mathbb{R}^d$ and $\mathbf{v}_t^i \in \mathbb{R}^d$ are position and velocity vectors respectively and $d$ is the dimensionality of the environment, typically 2 or 3. Let $\{\tau^i\}_{i=1}^N$, where $\tau^i = \mathbf{p}_{1:T}^i := (\mathbf{p}_1^i, \dots, \mathbf{p}_T^i)$, be the set of observed trajectories from an environment with $N$ entities. Together with a prior on $z$, the generative process of BSP defines a joint probability distribution $p(\mathcal{D}, z; F)$ over the observed trajectory data $\mathcal{D}$ and latent properties $z$ given the force function $F$.[1] An example of the generative process of a three-body problem is shown in figure 2.

The state transition of an entity in a Newtonian system depends not only on its properties and current state but also on its interaction with other entities in the environment. Therefore, we define a pairwise interaction function $F(z^i, \mathbf{s}^i, z^j, \mathbf{s}^j)$, where $i, j \in \{1 \dots N\}$; we interpret $F$ as the force applied to $i$ due to its interaction with $j$. Then, the trajectory $\tau_i$ of each entity is generated by a transition function $\mathbb{T}$ that consumes the current state and all of its interactions as

$$\mathbf{s}_{t+1}^i = \mathbb{T}\left(\mathbf{s}_t^i, F(z^i, \mathbf{s}_t^i, z^1, \mathbf{s}_t^1), \dots, F(z^i, \mathbf{s}_t^i, z^N, \mathbf{s}_t^N)\right). \tag{1}$$

As forces are additive, forces on entity $i$ can be easily summed to get the total force applied as $\mathbf{f}_t^i = \sum_{j=1}^N F(z^i, \mathbf{s}_t^i, z^j, \mathbf{s}_t^j)$. Similar to Sanchez-Gonzalez et al. (2019), we use numerical integration to simulate the Newtonian dynamics, updating $\mathbf{s}$ with the acceleration obtained from $\mathbf{f}_t^i$.

---

[1]As physical dynamics are typically sensitive to *initial states*, we assume the noise-free initial states are given either as part of the data $\mathcal{D}$ or as a point-mass prior over the initial state, thus are omitted in the notation.

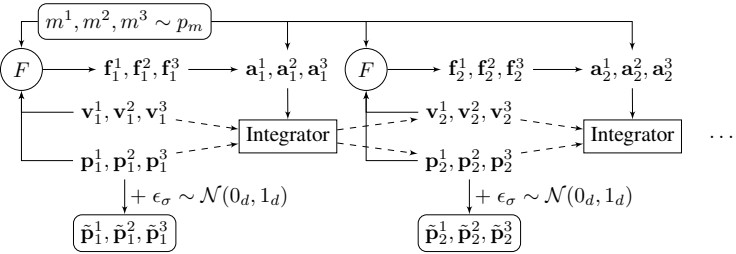

Figure 2: The generation of an observed trajectory: a three-body example with unknown mass. Circles are the learnable force function, rectangles are fixed functions, rounded rectangles are random variables and others are deterministic variables.

$$
\begin{aligned}
Constant &\rightarrow c_1 \mid c_2 \mid c_3 \\
Unitless &\rightarrow \mu_1 \mid \mu_2 \mid \mu_1 - \mu_2 \\
&\mid \mu_1 - \mu_2 \mid \mu_1 + \mu_2 \\
Kg &\rightarrow m_i \mid m_j \\
&\mid m_i - m_j \mid m_j + m_i \\
KgSq &\rightarrow m_i \times m_j \mid (Kg)^2 \\
MeterVec &\rightarrow \mathbf{p}_i - \mathbf{p}_j \\
&\mid \mathbf{p}_i - \mathbf{c} \mid \mathbf{p}_j - \mathbf{c} \\
MeterSecVec &\rightarrow \mathbf{v}_i \mid \mathbf{v}_j \mid \mathbf{v}_i - \mathbf{v}_j \\
Meter &\rightarrow \|MeterVec\|_2 \\
MeterSq &\rightarrow (Meter)^2 \\
MeterSec &\rightarrow \|MeterSecVec\|_2 \\
MeterSecSq &\rightarrow (MeterSec)^2
\end{aligned}
$$

$$
\begin{aligned}
RefInvVec &\rightarrow MeterVec \mid MeterSecVec \\
UnitlessVec &\rightarrow \text{normalize}(RefInvVec) \mid MeterVec \div Meter \\
&\mid MeterSecVec \div MeterSec \\
Meter &\rightarrow \text{project}(MeterVec, UnitlessVec) \\
MeterSec &\rightarrow \text{project}(MeterSecVec, UnitlessVec) \\
BaseCoeff &\rightarrow Unitless \mid Kg \mid KgSq \mid KgSq \div Kg \mid Meter \mid MeterSq \\
&\mid Meter - Meter \mid Meter + Meter \mid MeterSec \mid MeterSecSq \\
&\mid MeterSecSq + MeterSecSq \mid MeterSecSq - MeterSecSq \\
Coeff &\rightarrow BaseCoeff \mid BaseCoeff \times BaseCoeff \\
&\mid BaseCoeff \div BaseCoeff \\
BaseForce &\rightarrow Constant \times Coeff \times UnitlessVec \\
Bool &\rightarrow \text{isOn}(\mathbf{p}_i, s_i, \mathbf{p}_j, s_j) \mid \text{doesCollide}(\mathbf{p}_i, s_i, \mathbf{p}_j, s_j) \\
Force &\rightarrow BaseForce \mid BaseForce \times Bool \mid Force + BaseForce
\end{aligned}
$$

Figure 3: A grammar of expressions of Newtonian physical laws

Specifically, we choose the Euler integrator since its update rules correspond to the basic relations between position, velocity and acceleration. With these specifications, equation 1 becomes

$$
\mathbf{a}_t^i = \mathbf{f}_t^i / m^i, \quad \mathbf{v}_{t+1}^i = \mathbf{v}_t^i + \mathbf{a}_t \Delta t, \quad \mathbf{p}_{t+1}^i = \mathbf{p}_t^i + \mathbf{v}_{t+1}^i \Delta t, \tag{2}
$$

where $m^i$ is the mass of the recipient of the force $\mathbf{f}_t^i$ and $\Delta t$ is the step size of the Euler integrator. Finally, we add Gaussian noise to each trajectory $\{\tau^i\}_{i=1}^N$, that is, $\mathcal{D} := \{\tilde{\tau}^i\}_{i=1}^N$ where $\tilde{\tau}^i := (\tilde{\mathbf{p}}_1^i, \ldots, \tilde{\mathbf{p}}_T^i)$, $\tilde{\mathbf{p}}_t^i \sim \mathcal{N}(\mathbf{p}_t^i, \sigma^2)$ and $\sigma$ is the noise level. For clarity, Appendix A provides the complete generative process represented by a probabilistic program.

## 3.2 A GRAMMAR OF NEWTONIAN PHYSICS

In order attain sample efficiency, we chose to learn $F(z^i, \mathbf{s}^i, z^j, \mathbf{s}^j)$ using symbolic search, but this approach can be inefficient if the search space of possible functions is too large, or inaccurate if the search space is too small. Therefore, we constrain the function $F$ to be a members of a context-free language called the grammar of Newtonian physics, $\mathcal{G}$, which we now describe.

We consider the following terminal nodes in $\mathcal{G}$: learnable constants $c_1, c_2, c_3$, masses $m_i, m_j$, friction coefficients $\mu_i, \mu_j$, shapes $s_i, s_j$, positions $\mathbf{p}_i, \mathbf{p}_j$, velocities $\mathbf{v}_i, \mathbf{v}_j$ and the contact point $\mathbf{c}$ for a pair of entities.[2] and the following arithmetic expressions: $(\cdot)^2$ (square), $+$, $-$, $\times$, $\div$, $\|\cdot\|_2$ (L2-norm), $\text{normalise}(\cdot)$ and $\text{project}(\cdot, \cdot)$ (project a vector to an unit vector). However, naively supporting all possible arithmetic expressions for any combination of terminal nodes would make SR highly inefficient and even lead to *physically meaningless* force laws. Therefore, in order to constrain expression search further, we introduce physics-inspired production rules, along with preterminals and nonterminals nodes, as shown in Figure 3. We now discuss the design choice of our grammar.

Motivated by *dimensional analysis* in natural sciences, in which the relations between different units of measurement are tracked (Brescia, 2012), we build the concept of *units of measurement* into the nonterminals of $\mathcal{G}$. That is, mass has the unit kilogram ($Kg$), unit meter ($Meter$) for distance and meter per second ($MeterSec$) for speed. With this unit system in place, we only allow addition

---

[2]In cases of no contact, $\mathbf{c}$ is set as the middle position of the two objects, i.e. $\mathbf{c} = (\mathbf{p}_i + \mathbf{p}_j) / 2$.

---

**Algorithm 1:** Robust expectation–maximization for Bayesian-symbolic physics

---

initialize the force function $F_0$ as constantly zero ;

**for** $i = 1, \ldots, m$ **do**

    simulate $k + k'$ chains from $p(z \mid \mathcal{D}; F_{i-1})$ using HMC ;      // E-step starts

    compute ESS for each chain and remove $k'$ chains with the smallest ESS ;

    select the last sample from each chain as $\{z_1, \ldots, z_k\}$ ;

    get current loss function $\mathcal{L}_i(e, c) = \sum_{i=1}^{k} \mathcal{L}(e, c; z_i, \mathcal{D})$ ;      // M-step starts

    get candidates $\mathcal{C} = \{(t_1^*, c_1^*), \ldots, (t_r^*, c_r^*)\}$ by Algorithm 2 with $\mathcal{L}_i$ for $r$ repetitions ;

    find $(t^*, c^*)$ from $\mathcal{C}$ with the best loss and set $F_i = \text{getF}(t^*, c^*, \mathcal{G})$ ; // Update force

**end**

**return** $F = F_m$ and $\{z_1, \ldots, z_k\} \sim p(z \mid \mathcal{D}; F_m)$ ;

---

and subtraction of symbols with the same units, therefore pruning away physically meaningless expressions, e.g. $Kg - Meter$. Importantly, this leads to forces laws that have the unit of Newton.

We also forbid the direct use of *absolute* positions $\mathbf{p}_i$, $\mathbf{p}_j$ and $\mathbf{c}$ and only allow their differences.[3] This ensures that all force laws are *reference-invariant*, that is, independent on the choice of the reference frame. To be consistent with the unit system, we call vectors obtained from operations on positions vectors, *meter vector* ($MeterVec$) and those on velocities as *meter per second vector* ($MeterSecVec$). These variables are all vectors thus we call them reference-invariant vectors ($RefInvVec$). When such vectors are normalised, or divided by their corresponding "scalar variables", they becomes unit-less vectors ($UnitlessVec$) that can be used to describe a direction.

The start symbol of the grammar is $Force$. We allow forces to be summed by right-branching or be conditioned on a Boolean expression. In order to support *conditional forces*, which are forces that only apply when a condition is true, such as *collision force* and *friction*. Since the goal of our work is not perceptual learning, we provide perceptual primitives (collision detection as a function doesCollide and isOn to check if a disc is on a mat) in the grammar. Collision is then handled by applying an extra force to the entity when doesCollide is true; the force must still be learned.

Finally, some care is needed to ensure the grammar is unambiguous. For example, if we used a rule like $Coeff \rightarrow Coeff \times Coeff$, then the grammar could generate many expressions that redundantly represent the same function. This would make search much more expansive. Instead, we represent this rule in a right-branching way by the introduction of $BaseCoeff$ and $BaseForce$ as nonterminals. Although the grammar puts basic constraints on plausible physical laws, it is still expressive. For example, there are more than *6 million* possible trees of depth 6.

### 3.3 LEARNING ALGORITHM

Following the EM approach, our learning method (Algorithm 1) alternates between an E-step, where object properties are estimated given the current forces (Section 3.3.2), and an M-step step, where forces are learned given object properties (Section 3.3.1). See Appendix B.1 for hyperparameters.

### 3.3.1 SYMBOLIC REGRESSION WITH LEARNABLE CONSTANTS

Symbolic regression is function approximation that searches over the space of mathematical expressions that are specified by a user-provided context-free grammar (CFG; Koza, 1994). A CFG consists a start symbol, sets of nonterminal, preterminal and terminal symbols and a set of production rules. If each production rule in the grammar is specified with a probability (with the probabilities for all rules summed to 1), such a grammar is called probabilistic context-free grammars (PCFGs), which effectively defines a distribution over possible expression trees. As such, one can sample from a PCFG and/or evaluate the probability of a given tree. In our work, we use the cross-entropy method for SR. The method starts with PCFG of a given grammar that has equal probabilities of all production rules. At each iteration, it samples $n$ number of trees (up-to a specified depth $d$) from the current PCFG and evaluates their fitness by a loss function $\mathcal{L}$. After this, trees with the top-$k$ fitness are seleted and used to fit a PCFG, which will be used in the next iteration, via maximal likelihood.

---

[3]This is in fact consistent with how such variables are pre-processed in the neural network approaches. Usually the mean of a pair of positions are subtracted from the pair to make them reference-invariant.

---

**Algorithm 2:** Cross-entropy method with learnable constants

---

initialize a PCFG $\mathcal{P}_0$ for $\mathcal{G}$ uniformaly ;
**for** $i = 1, \ldots, m$ **do**
    initialize an empty candidate set $\mathcal{C}$ ;
    **for** $j = 1, \ldots, n$ **do**
        sample an expression $e_j \sim \mathcal{P}_{i-1}, e_{i-1}$ with a maximum depth of $d$ ;
        solve $c_j^* = \arg\min_c \mathcal{L}(e_j, c)$ by L-BFGS ;    // Lower-level optimization
        compute the loss of the sampled tree $\ell_j = \mathcal{L}(e_j, c_j^*)$ and add $(e_j, \ell_j)$ to $\mathcal{C}$ ;
    **end**
    **if** $i < m$ **then**
        fit a PCFG $\mathcal{P}_i$ on trees from $\mathcal{C}$ with the top-$k$ fitness via maximum-likelihood ;
**end**
**return** the best expression tree $e^*$ from $\mathcal{C}$ and the corresponding constant as $c^*$ ;

---

To learn the force laws, we need to find an expression $e \in \mathcal{G}$ and a setting for the learnable constants $c = [c_1, c_2, c_3]$ that define the force function $F_{e,c}$. The loss used by the cross-entropy method involves computing the log-likelihood of the generative model. As the observed trajectory is generated sequentially given an initial state, the computation of the log-likelihood term cannot be parallelized and can be computationally expensive in practice. Following the loss form of (Battaglia et al., 2016; Sanchez-Gonzalez et al., 2019), we use a *teacher-forced* or *vectorized* version of the log-likelihood

$$LL(e, c; z, \mathcal{D}) = \sum_{i=1}^{N} \sum_{t=1}^{T-1} \log \mathcal{N}(\tilde{\mathbf{p}}_{t+1}^i; \mathbb{T}\left(\tilde{\mathbf{s}}_t^i, F_{e,c}(z^i, \tilde{\mathbf{s}}_t^i, z^1, \tilde{\mathbf{s}}_t^1), \ldots, F_{e,c}(z^i, \tilde{\mathbf{s}}_t^i, z^N, \tilde{\mathbf{s}}_t^N)\right), \sigma) \quad (3)$$

where $\mathbb{T}$ follows equation 2 and $\tilde{\mathbf{s}}_t^i := (\tilde{\mathbf{p}}_t^i, \tilde{\mathbf{v}}_t^i)$. As such, we assume velocity is also available in the dataset $\mathcal{D}$. Clearly, equation 3 differs from the sequential version, as the input for the integrator contains noise at each step. However, similar to previous works, we found it is not an issue when learning forces by regression and allows a speed-up of 10x in terms of computing the log-likelihood.

In order to favor simpler trees, we add a regularization term, a weighted log-probability under a uniform PCFG prior of $\mathcal{G}$, to the negative log-likelihood; our final loss per trajectory

$$\mathcal{L}(e, c; z, \mathcal{D}) = -LL(e, c; z, \mathcal{D}) + \lambda \log \mathcal{P}_0(e). \quad (4)$$

Here $\mathcal{P}_0$ is the uniform PCFG of $\mathcal{G}$ and $\lambda$ is a hyper-parameter that controls the regularization. The loss for multiple trajectories is just a summation of $\mathcal{L}$ over individual trajectories. Optimizing equation 4 can be seen as *maximum a posterior* (MAP) estimation.

When using the cross-entropy method for symbolic regression, the continuous constants $c = [c_1, c_2, c_3]$ require care as they can take any value. To handle this, we use *bilevel optimization*, where the upper-level is the original symbolic regression problem and the inner-level is an extra optimization for constants. Specifically, we use an L-BFGS step to optimize the constants before computing the loss of each candidate tree within the cross-entropy iterations. For cases where the constants are very small, e.g. the gravitational constant $G = 6.67 \times 10^{-11}$, we parameterize the constants as $c \times 10^{-9}$ to avoid numerical issues in the inner-level optimization. Traditionally, if such strategy is not used, constants are either randomly generated from a predefined, fixed integer set or a continuous interval, or for evolutionary algorithms, they can be mutated and combined during evolution to produce constants that fit better; such constants are often referred as *ephemeral constants* (Davidson et al., 2001). Compared to these methods, the benefit of our formulation is that the evaluation of each tree candidate depends on the symbolic form *only* as the constants are *optimized-away*, making the search more efficiently. Note that although the literature has not explicitly considered our way of constant learning as bilevel optimization, such strategy is not new and is similar in spirit to (Cerny et al., 2008; Kommenda et al., 2013; Quade et al., 2016). In contrast to recent use of bilevel optimization in meta-learning, e.g. (Finn et al., 2017), our method is simpler: As our upper-level optimization is gradient-free, we do not need to pass gradient from the lower-level to the upper-level.

In practice, as the cross-entropy method itself is sensitive to random initializations, in order to robustify the M-step, we repeat it for $r$ runs and pick the best optimizer. We provide a complete description of cross-entropy method with learnable constants in Algorithm 2.

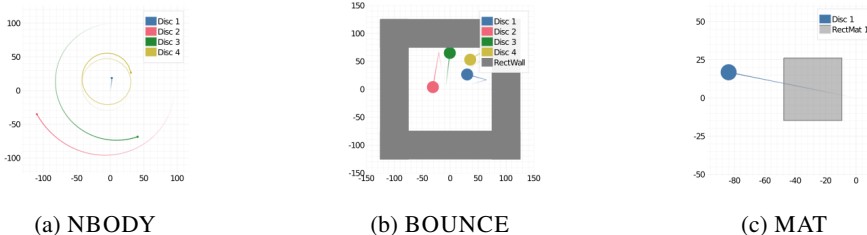

(a) NBODY        (b) BOUNCE        (c) MAT

Figure 4: Example scenes for the datasets used for evaluation. Entities in gray are static.

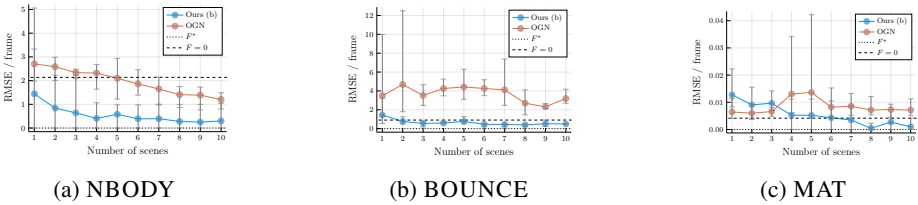

(a) NBODY        (b) BOUNCE        (c) MAT

Figure 5: Predictive error (RMSE) per frame on the held-out set with a varying number of scenes for training. The line plots are mean values out of five experiments with different shuffling of the training set and the error bars are minimum and maximum values.

### 3.3.2 REASONING ABOUT UNKNOWN PROPERTIES

With a force function $F$ given, reasoning of unknown properties is reduced to posterior inference on $z$ in the generative model specified in Section 3.1. Since for a fixed $F$, the model in our framework is end-to-end, piecewise differentiable with respect to properties, we perform inference by sampling using Hamiltonian Monte Carlo (HMC; Duane et al., 1987; Neal et al., 2011). Other particle-based alternatives like importance sampling and sequential Monte Carlo are possible but are less efficient.

In order to draw $k$ samples from the posterior *robustly* in the E-step, we first run $k + k'$ independent HMC chains by the no-U-turn sampler (NUTS; Hoffman & Gelman, 2011) for a reasonably large number of iterations, where $k'$ is also a hyper-parameter to choose. After this, we remove $k'$ chains with the smallest effective sample size (ESS). This reduces the chance of using samples from chains that mixed poorly or got stuck in bad region due to random initialization. Finally, we pick the last sample from each chain as the samples returned by the E-step $\{z_1, \ldots, z_k\}$.

## 4 EXPERIMENTS

In this section, we present a battery of empirical evaluations of the symbolic M-step and demonstrate how our complete EM algorithm is capable of joint reasoning and learning.

**Datasets.** We consider three simulated datasets for our evaluation. These three datasets, if combined together, correspond to the dataset used in Ullman et al. (2018) for assessing physics learning and reasoning in humans. The first dataset, NBODY (n-body simulation with 4 bodies), is populated by placing a heavy body with large mass and no velocity at $[0, 0]$ and three other bodies at random positions with random velocities such that, they would orbit the heavy body in the middle in the absence of the other two bodies. The gravitational constant is set such that the system is stable within the duration of the simulation. The ground truth force to learn is the gravitational force between the bodies. The second dataset, BOUNCE, is generated by simulating *elastic collision* between a set of discs and the box that they are contained within. The gravitational constant is set small such that the gravitational force is ignorable and the ground truth force to learn is the collision resolution force. The last dataset, MAT, simulates the friction-based interaction of discs and a mat. We populate this data by rolling discs with different initial states over mats with random sizes and positions and applying friction when they come in contact. The ground truth force to learn is this friction. Each dataset consists of 100 scenes, 20 of which are held-out for testing. All scenes are simulated using a physics engine with a discretization of 0.02, for 50 frames; see figure 4 for an example of each dataset and Appendix B.2 for the ground truth force expressions under our grammar.

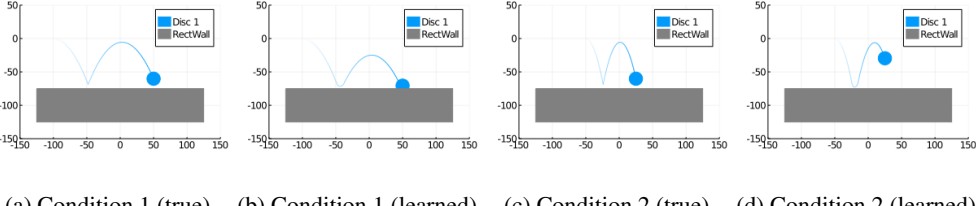

(a) Condition 1 (true)    (b) Condition 1 (learned)    (c) Condition 2 (true)    (d) Condition 2 (learned)

Figure 6: Generalization of the approximate bounce law in a vertical world with downward gravity.

**Baselines**  Being most closely related to our work, we use a specific instantiation of the OGN model (Sanchez-Gonzalez et al., 2019) as the neural baseline. The original OGN model does not assume any particular Hamiltonian dynamics and thus outputs the partial derivatives of both position and momentum variables to integrate the dynamics. In our specific realization, we only make it to output the partial derivative of the velocity variable since under the Newtonian dynamics, the partial derivative of the position variable is analytically known as the velocity. Since OGN (and other neural methods, such as IN) is designed to learn with all properties given, we assume all the properties are fully observed and compare our symbolic force model against OGN. See Appendix B.3 for details of the neural architecture, training and parameterization setup for OGN.

## 4.1  DATA-EFFICIENCY: SYMBOLIC VS NEURAL

We now compare the symbolic M-step of BSP against the OGN-based neural baselines in terms of data-efficiency. Specifically, we check how accurate the per-frame predictions of these models are on held-out data, and how their accuracies change with the amount of training data. We use a *noise-free* version of the trajectory in this evaluation and provide all the properties as observed data since the neural baselines cannot be trained if the properties are not fully observed. For each dataset, holding out 20 scenes for evaluation, we randomly shuffle the remaining 80 scenes for training and use the first $k$ scenes to fit the model. Because an average human can perform such learning tasks with about 5 scenes (Ullman et al., 2018), we only vary $k$ from 1 to 10 in our experiments. We use the root mean squared error (RMSE) per frame as the performance metric, repeat each of the experiments five times with different training set and finally report the mean, maximum and minimum performance for all the methods, as shown in figure 5. As reference, we include the performance of $F = 0$, a zero-force baseline, corresponding to the constant velocity baseline in (Battaglia et al., 2016). We also include the performance of the ground truth force $F^*$, which has an RMSE per frame of 0. As it can be seen, the symbolic M-step is more sample-efficient than the neural baseline for all datasets within this limited data regime. For NBODY and MAT, BSP can find the ground truth force function with 1 scene and 4 scenes respectively. For BOUNCE, the neural baseline cannot reach the performance of $F = 0$ even after 10 scenes for training. This is a known issue with neural network approaches when learning collision as the inherently sparse nature of the collision interaction does not provide enough training signal (Battaglia et al., 2016). As BOUNCE is the only case where our method fails to find the true law within 10 scenes, we include the typical inferred law in Appendix C as well the predicated trajectories of some selected scenes for inspection.

## 4.2  GENERALIZATION

It is worth checking how the laws learned in Section 4.1 generalize to new scenes beyond the training data. In cases where the true law is successfully recovered, the expression will generalize to novel scenes undoubtedly. Therefore, it is more interesting to inspect the generalization ability of an approximate law, that is a law which is not completely equivalent to the true law but is close. The emerged law for the BOUNCE dataset is such an example as mentioned earlier. It has an expression of $F^\dagger = c \, \|\mathbf{v}_i - \mathbf{v}_j\|_2 \, \frac{\mathbf{p}_i - \mathbf{c}}{\|\mathbf{p}_i - \mathbf{c}\|_2} \, \text{doesCollide}(\mathbf{p}_i, s_i, \mathbf{p}_j, s_j)$; see figure 10 in Appendix C for the actual tree. Although it is not identical to the true law, it is still a good approximation: it takes into accounts the velocity difference into consideration and finds the correct force direction. We now consider applying this law to a completely new scene: a vertical-view world where the gravity is pointing in downward direction. figure 6 shows the predicted trajectory with true and the approximate law with two different initial conditions. As it can be seen, the approximate law successfully generalize this novel world. For the first condition, the projection is very close to the true one, while

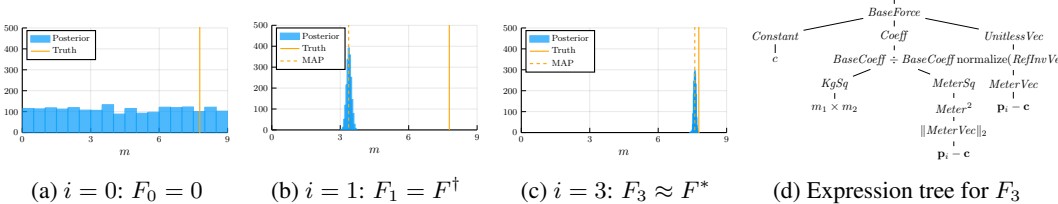

(a) $i = 0$: $F_0 = 0$     (b) $i = 1$: $F_1 = F^\dagger$     (c) $i = 3$: $F_3 \approx F^*$     (d) Expression tree for $F_3$

Figure 7: Results of the EM algorithm on NBODY. figure 7a to figure 7c shows the posterior of mass for Entity 1 in Scene 1 with the corresponding force function during EM. In figure 7b, the force function $F^\dagger = 239.99 \frac{m_i m_j}{\|\mathbf{p}_i - \mathbf{p}_i\|_2} \frac{\mathbf{p}_i - \mathbf{p}_j}{\|\mathbf{p}_i - \mathbf{p}_j\|_2}$. The constant in figure 7d is $c = 2.04 \times 10^3$.

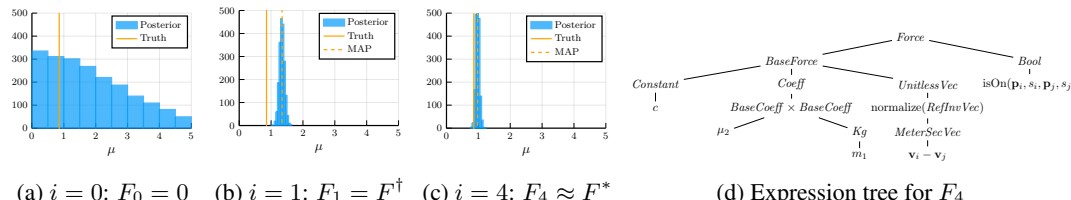

(a) $i = 0$: $F_0 = 0$    (b) $i = 1$: $F_1 = F^\dagger$    (c) $i = 4$: $F_4 \approx F^*$     (d) Expression tree for $F_4$

Figure 8: Results of the EM algorithm on MAT. figure 8a to figure 8c shows the posterior of friction coefficient in Scene 2 with the corresponding force function during EM. In figure 8b, the force function $F^\dagger = -22.99 \, \mu_j \frac{\mathbf{v}_i}{\|\mathbf{v}_i\|_2} \, \text{isOn}(\mathbf{p}_i, s_i, \mathbf{p}_j, s_j)$. The constant in figure 8d is $c = -8.605$.

for the second condition, the concept of bounce is also correctly transferred. The corresponding animations for these plots can also be found in the supplementary material for further inspections.

### 4.3 LEARNING WITH UNOBSERVED PROPERTIES

Now, we demonstrate how the joint learning and reasoning in the BSP method (Algorithm 1) can recover the true force law when some properties are unobserved. As the first experiment, we use three scenes from the (noisy) NBODY dataset (with four entities per scene), such that if the true masses are given, the M-step can successfully learn the true force law. Next, we assume that the masses of the three light entities are unknown with a uniform prior $\mathcal{U}(0.02, 9)$ and the mass of the heavy entity is known. We use Algorithm 1 to fit the same generative model that simulates the data using BSP. figure 7 shows the posterior distribution over mass and the force function at initialization (figure 7a), middle (figure 7b) and convergence (figure 7c). In this run, after 3 iterations, our algorithm successfully recovers the true force function. We repeat this experiment ten times with randomly sampled scenes and for eight of them, BSP successfully recovers the true force law. Note that because the intermediate learned force law $F^\dagger$ is incorrect, the variance of the posterior (in figure 7b) is larger than the one from the true force law (in figure 7c). Compared the expression at convergence in figure 8d with the true law, the algorithm replaces $\mathbf{p}_i - \mathbf{p}_j$ with $\mathbf{p}_i - \mathbf{c}$ and a scaled constant. This is valid as the contact point is defined as $\mathbf{c} = (\mathbf{p}_i + \mathbf{p}_j)/2$ when there is no contact.

For the second experiment, we use five scenes from the (noisy) MAT dataset. We assume that the only unknown is the friction coefficient of the mat with a truncated Gaussian prior $\mathcal{T}\text{runcated}(\mathcal{N}(\mu_0, 2^2), 0, 5)$ (truncated between 0 and 5), where $\mu_0$ is the true coefficient, Note that the variance $2^2$ is large enough to be uncertain, justifying a fair choice of the prior. Similarly, we use Algorithm 1 to fit the same generative model that simulates the data using BSP. figure 8 shows the posterior distribution over mass and the force function at initialization (8a), middle (8b) and convergence (8c) of the algorithm. Compared the expression at convergence with the true law, the algorithm learns $\mathbf{v}_i - \mathbf{v}_j$ instead of $\mathbf{v}_i$ as the mat velocity is zero, i.e. $\mathbf{v}_j = 0$, in all scenes,

## 5 DISCUSSION AND CONCLUSION

We present BSP, the first fully Bayesian approach to symbolic intuitive physics by combining symbolic learning of physical force laws and statistical learning of unobserved properties. Our work paves the way for using learnable, data-efficient IPEs in intuitive physics by providing a computational framework to study how humans' iterative reasoning-learning is mentally performed.

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

---

**Algorithm 3:** Complete generative process given force laws

---

```
// Sample latent variables
```
**for** $i = 1, \ldots, N$ **do**
    ASSUME $z^i$ from prior for entity $i$ ;
    **if** *initial state is not given* **then**
        ASSUME $\mathbf{p}_0^i$ from prior for entity $i$ ;
        ASSUME $\mathbf{v}_0^i$ from prior for entity $i$ ;
    set $\mathbf{s}_0^i = (\mathbf{p}_0^i, \mathbf{v}_0^i)$ ;
**end**
**for** $t = 1, \ldots, T$ **do**
    **for** $i = 1, \ldots, N$ **do**
        `// Compute force and acceleration`
        **for** $j = 1, \ldots, N$ **do**
            compute $\mathbf{f}_t^{i,j} = F(z^i, \mathbf{s}_{t-1}^i, z^j, \mathbf{s}_{t-1}^i)$ ;
        **end**
        compute $\mathbf{f}_t^i = \sum_{j=1}^N \mathbf{f}_t^{i,j}$ ;
        compute $\mathbf{a}_t^i = \mathbf{f}_t^i / m^i$ ;
        `// Euler's integration`
        update $\mathbf{v}_t^i = \mathbf{v}_{t-1}^i + \mathbf{a}_t \Delta t$ ;
        update $\mathbf{p}_t^i = \mathbf{p}_{t-1}^i + \mathbf{v}_t^i \Delta t$ ;
        set $\mathbf{s}_t^i = (\mathbf{p}_t^i, \mathbf{v}_t^i)$ ;
        `// Sample observations`
        OBSERVE $\tilde{\mathbf{p}}_t^i$ from $\mathcal{N}(\mathbf{p}_t^i, \sigma^2)$ ;
    **end**
**end**

---

## A    COMPLETE GENERATIVE PROCESS

Section 3 describes the top-down generative model piece by piece, to improve the clarity of the EM framework, we provide the complete generative process of the observation given force function $F$, which corresponds to the E-step in our method, as a probabilistic program in Algorithm 3. In this probabilistic program, we use the keyword ASSUME and OBSERVE for sampling latent variables and observations separately, following the notations from Wood et al. (2014).

## B    EXPERIMENTAL DETAILS

### B.1    HYPER-PARAMETERS FOR ALGORITHM 1

For the E-step, we use $k = 3$ and $k' = 2$ and the hyper-parameters for NUTS are: $150$ adaptation steps, $150$ HMC iterations, a maximum tree depth of $4$ and a target acceptance ratio of $0.75$. For the M-step, we repeat $r = 2$ runs and the hyper-parameters for the cross-entropy method are: $1,000$ total populations, $500$ selected populations, $4$ iterations and a maximum depth of $8$. The weighting parameter for the PCFG prior are $1$ for the NBODY and BOUNCE datasets and $1 \times 10^{-4}$ for the MAT dataset.

### B.2    GROUND TRUTH FORCES

The symbolic tress of ground truth forces that are used to generated the datasets used in Section 4 are given in figure 9.

### B.3    TRAINING SETUP OF OGNS

For the OGN baseline, we use a multilayer perceptron (MLP) of $d_{\text{in}} \to 50 \to 50$ where the activation function is the rectified linear unit (ReLU) for the node model and use a MLP of $(50 + 50) \to 50 \to 50 \to 50 \to d_{\text{out}}$ as the edge model. For training, we use the ADAM optimizer (Kingma

& Ba, 2014) with a learning rate of $5 \times 10^{-3}$ for a total number of 10,000 passes of scenes for the NBODY and MAT datasets and a total number of 20,000 passes of scenes for the BOUNCE dataset. For example, if 5 scenes are used, the total number of pass of the dataset would be $10000/5$, for the NBODY dataset. This makes sure that the training time for all experiments are fixed.

In addition, we found that we need provide additional prior knowledge on how forces are related to the mass and acceleration by parameterzing them as $F_e(\cdot) = m\, a_\theta(\cdot)$, where $\theta$ is NN parameters, otherwise they fail to learn. This parameterization is fact consistent with (Sanchez-Gonzalez et al., 2019) in which NNs output partial derivatives of the Hamiltonian system.

## C   THE LEARNED BOUNCE LAW

As mentioned in Section 4.1 and discussed in Section 4.2, the only case in which BSP fails to infer the true law (within 10 scenes) is of special interest for further inspection. A typical approximate law learned in Section 4.1 is shown in figure 10; see Section 4.2 for discussion on how this law differs from the true one. To highlight, there are basically two mismatches between the true law and the learned law. First, there is no projection operation that correctly calculates the effect of speed. Second, the mass-based coefficient is missing. To assist inspection, we also provide some visualizations in figure 11 using initial conditions from the training set for inspection. The corresponding animations can be found in the supplementary material.

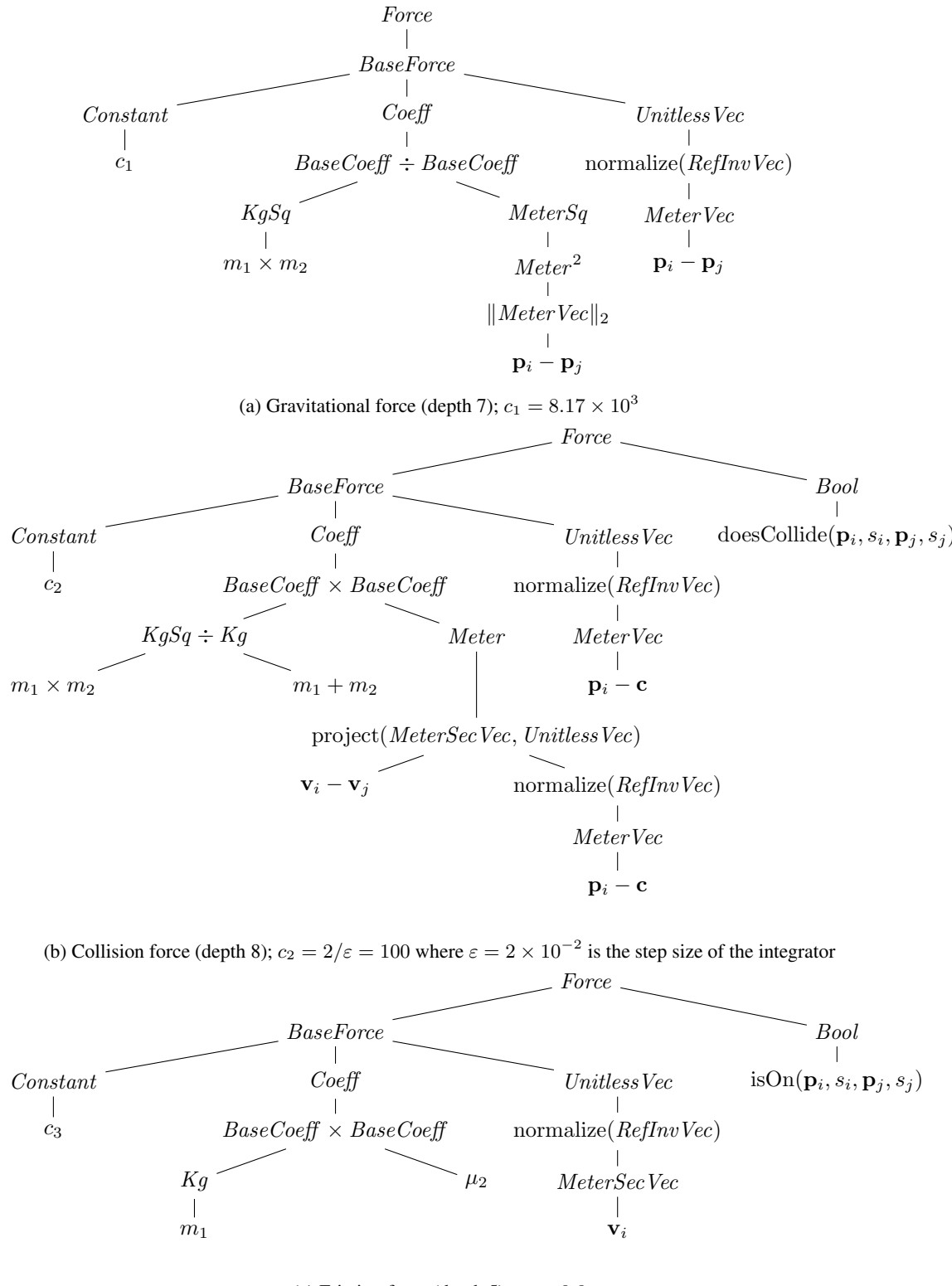

(a) Gravitational force (depth 7); $c_1 = 8.17 \times 10^3$

(b) Collision force (depth 8); $c_2 = 2/\varepsilon = 100$ where $\varepsilon = 2 \times 10^{-2}$ is the step size of the integrator

(c) Friction force (depth 5); $c_3 = 9.8$

Figure 9: Expression trees (under $\mathcal{G}$) of true force laws that generates the datasets used in Section 4.

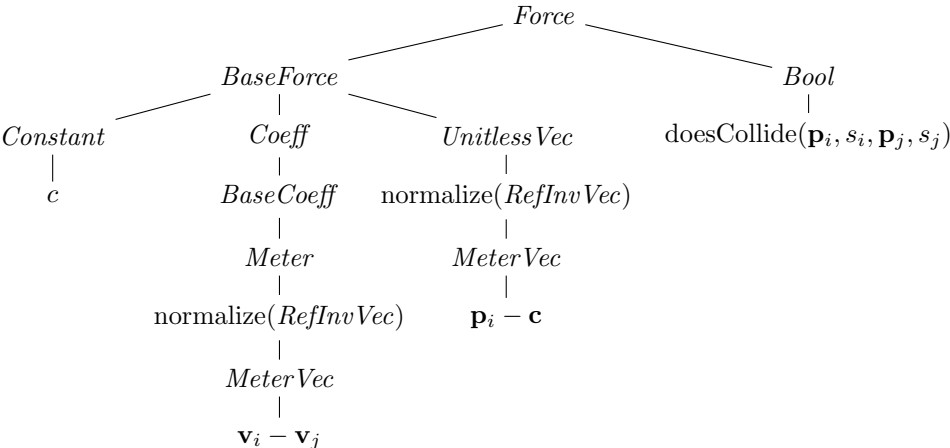

Figure 10: Approximate bounce law learned by BSP under our grammar; $c = 130.22$

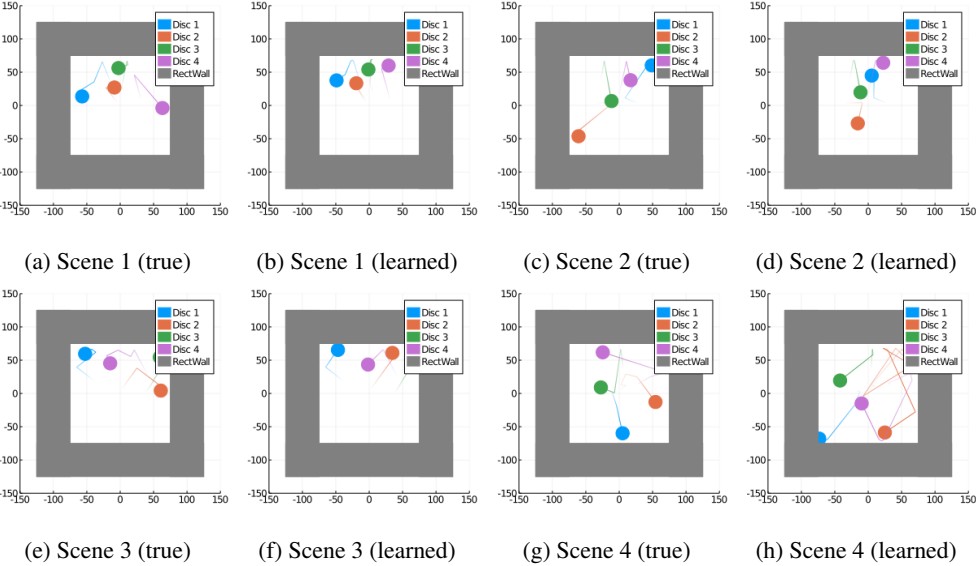

(a) Scene 1 (true)     (b) Scene 1 (learned)     (c) Scene 2 (true)     (d) Scene 2 (learned)

(e) Scene 3 (true)     (f) Scene 3 (learned)     (g) Scene 4 (true)     (h) Scene 4 (learned)

Figure 11: Predicated trajectories of the true bounce law and the learned bounce law.

