# OpenReview forum: "A Bayesian-Symbolic Approach to Learning and Reasoning for Intuitive Physics"
_ICLR.cc/2021/Conference — Reject_

### Official Review · AnonReviewer3 · 2020-10-26
**interesting position in the spectrum of intuitive physics models**

**Rating:** 6
**Confidence:** 4

**Review:**

The paper proposes an Bayesian-symbolic physics (BSP), an intuitive physics model that jointly infers symbolic force laws and object properties (mass, friction coefficient). The inductive bias is force summation, F=ma, and a grammar of force laws to express object interactions. The inference is done via an EM method that alternates between object property estimation (E-step) and force law induction (M-step), using techniques like symbolic regression and Hamiltonian Monte Carlo (HMC). Some preliminary experiments are shown for the method's effectiveness and data efficiency.

**Strength**:
The paper fills an important missing position in the spectrum of intuitive physics models, as Figure 1 argues. The force law grammar, to my knowledge, is something novel in this area, and represents a reasonable inductive bias that balances expressivity and physical plausibility (with two further physical constraints: dimensional analysis and reference invariance). The grammar also helps improve data efficiency. From the inference side, the proposed EM approach is also reasonable. Symbolic regression for force law inference can be interesting for future research.

**Weakness**:
Experiments seem a bit weak for now, and I have some technical concerns about the inference.

**Questions/suggestions**:
1. Learning with unobserved properties (Figure 6) is the key experiment setting according to the paper's selling point, but is obviously lacking. It'd be great to see more scenarios (than graviton) with more diverse settings (e.g. object mass, initial position), some quantitive numbers, and comparison with some baseline (if possible).
2. For some qualitative sense, would also be nice to show true trajectories vs. predicted trajectories in the main paper, or if the learned symbolic law force is correct (for Figure 5). Would be interesting to see if any symbolic laws are predicted wrong and how they look like.
3. For object property inference, why use Monte Carlo methods over variational/gradient-based methods? Also, is it possible to use MCMC for formula inference? Some explanations or experiments would solidify design choices for the inference part.
4. Is there an ambiguity when force constants and object properties are jointly inferred? How tricky is designing priors for these? (I see some tricks are used in the paper to handle small constants, for example)
5. The paper currently only compares with one side of the spectrum, i.e. more neural approaches. Comparing BSP with more symbolic approaches like (Smith et al., 2019), the strength is supposed to be the ability to handle non-standard object interactions outside fixed physics engine but within the force grammar. Is it possible to generate some random force laws and show BSP still works while baselines from BOTH sides of the spectrum may fail? It could be the most powerful experiment to support this position in the spectrum.

In general I like the ideas and hope to see more experiments and justifications for design choices.

---

> ### Author Response · Authors · 2020-11-16
> **Thanks a lot for your feedback and for acknowledging the gap that our work fills.**
>
> Re. *"Learning with unobserved properties (Figure 6) is the key experiment setting according to the paper's selling point, but is obviously lacking. I'd be great to see more scenarios (than gravitation)."*
>
> Thanks for the suggestion. We added one more experiment for the complete EM algorithm in the revised draft. We show how the EM algorithm can be applied to carry out inference in the MAT scenario.
>
> Re. *"For some qualitative sense, would also be nice to show true trajectories vs. predicted trajectories in the main paper ... Would be interesting to see if any symbolic laws are predicted wrong and how they look like."*
>
> Thanks for this suggestion, we agree such qualitative results would be helpful to understand the method. For the NBODY and MAT dataset, the exact true laws are found by the method, so such results might not be interesting. But for the BOUNCE one, in which the method failed to learn the exact ground-truth law, the results are indeed very interesting. The force law it learns is: `F = c * norm(v_i - v_j) * normalize(p_i - contact) * does_collide(s_i, p_i, s_j, p_j)`. We provide four pairs of true-vs-approximate comparisons in Appendix C along with the expression tree for this approximate law (the true one can be found in the appendix as well). As the difference is better to see in animations, we also provide the animated version in the `bounce_inspection` folder of the supplementary material.
>
> As mentioned, the approximate force is not the same as the true one. In comparison, the learned law basically misses two aspects (1) the projection operation that correctly calculates the effect of speed and (2) the mass-based coefficient. Nevertheless, the learned law is still a very good approximation to the true one: it takes into account the velocity difference, and finds the correct force direction, as indicated by the predicted trajectory. Interestingly, this approximate law can be used in a completely new scenario as well, shown by our new generalisation result in Section 4.2.
>
> Re. *"For object property inference, why use Monte Carlo methods over variational/gradient-based methods? ... Some explanations or experiments would solidify design choices for the inference part."*
>
> While we have not tried, Variational Inference (VI) should be possible. Though, VI in general,  would be (1) either limited by the choice of variational approximations or (2) if we use neural network based methods for flexible approximations, it's computationally very expensive. Overall, we think VI adds another layer of complexity, which we want to avoid. Besides, HMC also uses the underlying gradients and we found it sufficiently fast for the scenarios considered in this work. In principle, other particle-based MCMC methods can be used as well in the E-step on the expense of making it less efficient.
>
> Re. *"Also, is it possible to use MCMC for formula inference?"*
>
> Yes it is possible. MCMC algorithms for symbolic regression have been used in similar contexts before, e.g. [1], and related methods have recently been developed further [2]. We believe making our method fully Bayesian is very interesting (both methodology-wise and application-wise) as it allows joint exploration of force laws and properties. We are happy to further discuss this if the reviewer has some more thoughts on it.
>
> Re. *"Is there an ambiguity when force constants and object properties are jointly inferred? How tricky is designing priors for these? (I see some tricks are used in the paper to handle small constants, for example)"*
>
> Yes there will be ambiguity in a lot of cases if we jointly infer the constants and properties. In some cases, using more scenes will serve for disambiguation (NBODY) while for some cases this might not be helpful (MAT). In general, we do believe for the joint case, humans need a strong prior to perform sensible learning and inference tasks, so does our EM setup.
>
> Regarding the trick for some constants: it is mainly to deal with numerical issues in SR. If the constant is too small in scale, the inner optimization within the cross-entropy method can be problematic because they rely on iterative methods with some level of tolerance for termination.
>
> Re. *"The paper currently only compares with one side of the spectrum, i.e. more neural approaches … Is it possible to generate some random force laws and show BSP still works while baselines from BOTH sides of the spectrum may fail? It could be the most powerful experiment to support this position in the spectrum."*
>
> We agree that a fixed physics engine will fail in the suggested setup but such failure cases are also very expected. We therefore do not present experiments for such but we are happy to add some more discussions on the failure modes of fixed physics in our paper.
>
> ---
>
> [1] Goodman, Noah D., et al. "A rational analysis of rule‐based concept learning." Cognitive science 32.1 (2008): 108-154.
>
> [2] Jin, Ying, et al. "Bayesian Symbolic Regression." arXiv:1910.08892 (2019).

---

### Official Review · AnonReviewer2 · 2020-10-29
**A nice Bayesian approach to learn an intuitive physics model, but the relations with prior work and more comprehensive results are missing.**

**Rating:** 6
**Confidence:** 4

**Review:**

This paper proposes a fully Bayesian approach to learn an intuitive physics model by combining symbolic regression and statistical learning (MCMC).

Pros:

+ Personally, I like this paper as it approaches the learning of the intuitive physics model in the right direction, highlighted in figure 1. It is a nice combination of symbolic and learning-based approach.

Cons:

- The related work section is too brief to see the main difference between the present work and prior work. The current related work section only focuses on the machine learning-based intuitive physics model but does not cover symbolic regression in general. Using symbolic regression has a long history, especially in material science, soft robotics, and machine learning in general. Prior work has demonstrated that SR can indeed learn the physical law in a much more complex setting [1-2].

- For researchers who are opposed to the idea of intuitive physics, they would ask where the prior knowledge comes from. Some cognitive research has shown that they are innate for humans and some animals. But this rule cannot be applied for a machine: If we give them all the rules, why not just directly use the physics engine? In the experiments, the results demonstrated here are too simple so that most of the naive physics-based simulator can produce similar results at a probably faster speed. The question is, what the benefit of learning? One may argue that if I know some properties individually, I might be able to transfer this knowledge to unseen scenarios. The authors do demonstrate some capability in 3.3.2 and 4.2, but it seems to come naturally with SR based on the given prior, not something new or surprise. Such capabilities have been demonstrated in [1-2].

- I would like to see if the authors would be able to demonstrate the learned physics can be generalized to scenarios beyond the training datasets. For instance, can the learned model from MAT dataset be transferred to simple scenarios created by bullet-like engine with a similar friction-based interaction, but not identical? The learned physical knowledge should be general enough. Even not as the perfectly correct Newtonian physics, it should be able to generalize to unseen scenarios.

[1] Distilling free-form natural laws from experimental data, Science 2009
[2] AI Feynman: A physics-inspired method for symbolic regression, Science Advance 2020

---

> ### Author Response · Authors · 2020-11-16
> **Thanks a lot for the comments and suggestions and for acknowledging the importance of this direction.**
>
> Re. *"The related work section is too brief to see the main difference between the present work and prior work ... Prior work has demonstrated that SR can indeed learn the physical law in a much more complex setting [1-2]."*
>
> Thanks for the references, we have added them to the manuscript along with other relevant SR based work at the end of the related work section (Section 2). Please see the next response for how BSP differs from [1,2].
>
> Re. *"If we give them all the rules, why not just directly use the physics engine? In the experiments, the results demonstrated here are too simple so that most of the naive physics-based simulator can produce similar results at a probably faster speed. The question is, what the benefit of learning? One may argue that if I know some properties individually, I might be able to transfer this knowledge to unseen scenarios. The authors do demonstrate some capability in 3.3.2 and 4.2, but it seems to come naturally with SR based on the given prior, not something new or surprise."*
>
> Our grammar does not provide rules, rather it defines a search space of possible ways the world could work, with more than 6 million expression trees (at maximum depth of 6). Therefore, our model is not equivalent to directly using a physics engine, but to a search for the best physics engine. The goal of our work is to demonstrate that under the correct prior knowledge, rules for intuitive physics prediction can be learned in a data efficient manner, as is believed to happen in humans.
>
> Further, often, intuitive physics based predictions (such as B,C and D in [3]) are different (and incorrect) from what the laws of physics would predict. This is why it is important to enable learning in such models to discover how these laws of intuitive physics differ from that of general physics. In other words, our framework enables the possibility of mimicking human mistakes, which is very important in understanding intuitive physics reasoning. Our framework supports future research in which typical human errors and misconceptions across our physics-environments can be linked back to human physical priors (early developing or innate intuitive physical knowledge).
>
> Finally, an important focus of our work is to enable simultaneous learning of physical laws and inference of unknown physical properties in a data efficient manner. Our Bayesian approach is fairly different from [1,2] that aim to learn the physical equations (that directly leads to the observed data but unlike our approach) but do not support inference of unknown properties in physical environments, or data efficiency. In fact, [2] also uses neural networks that require quite a lot of data.
>
> Re. *"I would like to see if the authors would be able to demonstrate the learned physics can be generalized to scenarios beyond the training datasets. For instance, can the learned model from MAT dataset be transferred to simple scenarios created by bullet-like engine with a similar friction-based interaction, but not identical? The learned physical knowledge should be general enough. Even not as the perfectly correct Newtonian physics, it should be able to generalize to unseen scenarios."*
>
> Thanks for your suggestion - we have added a new experiment based on it. Please see point 1 of the Summary Response for a summary or the new Section 4.2 (titled "Generalization") in our revised paper for details.
>
> ---
>
> [1] Distilling free-form natural laws from experimental data, Science 2009
>
> [2] AI Feynman: A physics-inspired method for symbolic regression, Science Advance 2020
>
> [3] Kubricht, James R., Keith J. Holyoak, and Hongjing Lu. "Intuitive physics: Current research and controversies." Trends in cognitive sciences 21.10 (2017): 749-759.

---

### Official Review · AnonReviewer1 · 2020-10-29
**A computational symbolic framework for intuitive physics**

**Rating:** 6
**Confidence:** 3

**Review:**

The paper proposes a fully Bayesian approach to symbolic intuitive physics that, by combining symbolic learning of physical force laws and statistical learning of unobserved properties of objects. The paper proposes an EM-based method where in E-step object properties distribution are sampled using the current best estimated force laws and in M-step symbolic regression is used to update those laws.

The paper is clearly-written with clear explanation of the proposed EM-style method. However, one of the main claim that the method "enjoys the sample efficiency of symbolic methods with the ac- curacy and generalization of data-driven learned approaches" is not well-supported in the experiments. While in Sec. 4.1 it shows the proposed method is more data efficient than the neural baseline, it's not clear how the method generalized to complete different scenarios. Ideally the symbolic physical laws are universal thus can be applied to all physical interactions.

Some other questions / comments:
- One of the contribution states "Through empirical evaluations, we demonstrate that the BSP approach reaches human-like sample efficiency, often just requiring 1 to 5 observations to learn the exact force laws – usually more than 10x fewer than that of the closest neural alternatives." This (10x more data-efficient) is hard to tell from figure 5 as it doesn't show when the neural baseline reaches the same performance.
- The error bars in figure 5 is somewhat not very indicative of the stability of the method and some of them for the neural baseline are extremely large. It states that  the values are "out of five experiments with different shuffling of the training set". How many different shuffling is there?
- Sec 3.3.1 "In practice, as the cross-entropy method itself is sensitive to random initializations, in order to ro- bustify the M-step, we repeat it for r runs and pick the best optimizer." Is 'm' supposed to be r in Algorithm 2?
- For figure 6, if it converges after iteration 3, can we show the expression tree for F3 instead F5? How much do they differ?

---

> ### Author Response · Authors · 2020-11-16
> **We thank the reviewer for the helpful comments and suggestions!**
>
> Re. *"While in Sec. 4.1 it shows the proposed method is more data efficient than the neural baseline, it's not clear how the method generalized to complete different scenarios. Ideally the symbolic physical laws are universal thus can be applied to all physical interactions."*
>
> Thanks for the good point - we have added a new experiment based on it. Please see point 1 of the Summary Response for a summary or the new Section 4.2 (titled "Generalization") in our revised paper for details.
>
> Re. *"The error bars in figure 5 is somewhat not very indicative of the stability of the method and some of them for the neural baseline are extremely large. It states that the values are "out of five experiments with different shuffling of the training set". How many different shuffling is there?"*
>
> We run each experiment scenario 5 times and at the starting of each run, the dataset is shuffled. When the dataset is very small, the neural network overfits to the training set, leading to the high variance. To be more precise, each time, we shuffle 100 scenes and pick the top-$k$ ($k=1, \dots ,10$) scenes. The high variance comes from the fact that the $k$ scenes may not be diverse/good enough to provide sufficient training signal.
>
> Re. *"This (10x more data-efficient) is hard to tell from figure 5 as it doesn't show when the neural baseline reaches the same performance."*
>
> As explained in the previous question, the performance of the neural baseline can have high variance depending on the quality of the K random training scenes. Therefore, we used the best y-value for each of the x-values in Figure 5 in this comparison. For both, NBODY and BOUNCE, the best values for the neural baseline after 10 scenes is worse than that of ours with only 1 scene. So at least, in these two cases, our model is more than 10x data-efficient. We are happy to tweak the claim to make it more precise.
>
> Re. *"Is 'm' supposed to be r in Algorithm 2?"*
>
> Algorithm 2 only describes a single run of the M-step, in which $m$ is the number of cross-entropy iterations and $n$ is the number of population. The repetition number $r$ of M-step is in the complete EM procedure as in Algorithm 1.
>
> Re. *"For figure 6, if it converges after iteration 3, can we show the expression tree for F3 instead F5? How much do they differ?"*
>
> Sorry for the confusion. This is actually a typo. The tree is supposed to be for iteration 3, i.e. it is $F_3$; we have it fixed now in the revised draft. The expression is stable as long as the M-step is robust, which is the reason we repeat it for $r$ runs in the complete EM.

---

### Official Review · AnonReviewer4 · 2020-11-01
**The proposed grammar is claimed to be generic, however its generality is still limited**

**Rating:** 5
**Confidence:** 4

**Review:**

The paper addresses the problem of sample-efficient inference for symbolic physical rules. In the literature, there exists neural-network based models for learning a physical engine which have good predictive accuracy but poor sample efficiency, as well as symbolic models which are highly sensitive to deviations from their fixed physics engine. To be able to overcome issues as such, authors propose a generative model along with a symbolic regression framework, in which forces are produced from a probabilistic context free grammar that is designed to mimic simple Newtonian physics. This particular grammar is parameterized by a few latent variables related to unobserved properties of the physical environment, such as mass and charge. Finally, they develop an Expectation-Maximization algorithm, in order for estimating these latent variables as well as inferring the underlying physical laws of the system.

The methodology described in the paper enables prior physical knowledge to be incorporated into the statistical machine learning models in the form of a probabilistic context free grammar, which in return makes inference via EM applicable. The paper is well written in general, i.e. the main idea of the paper is easy to grasp and all the related technical concepts are explained in a very simple way. Mathematical notation is clear and consistent. Proposed methodology is novel and sound. Claims of the authors are supported by the experimental results on simulated datasets, and there seems to be no fallacy in empirical evaluations. Nevertheless, experimental section can be extended by large-scale applications or real world datasets. In other words, learning the underlying physical rules from a data set which is collected through real-world sensors would provide further evidence to the impact of such a framework. In addition, I think it’s necessary to include more results from symbolic and neural-network based learned models as a reference. Although relation to prior work is clearly addressed in the paper, authors compare their results to only OGN.

In my opinion, learning symbolic rules for physical reasoning and prediction in a sample-efficient manner will certainly be of interest to the ICLR community. I believe that the proposed methodology is clearly worth exploring, and the presented experimental results are promising. On the other hand, it is no surprise that a carefully constructed grammar for physics is sample-efficient. Authors chose to restrict the grammar with a small set of production rules, in order to narrow down the search space for deriving force terms. However, this can be also seen as a compromise from the generality of the model, since such a predefined grammar might be too restrictive and it might necessitate adding new rules for each new physical phenomena. For instance, in the fields related to optics or thermodynamics, to be able to model phenomena such as refraction-diffraction of light or heat transfer, one might need to modify the grammar manually.

Another drawback of the methodology could be enforcing position and velocity vectors to reside in the Cartesian coordinate system. Such a choice limits the expressiveness of the model, for instance allowing a general coordinate system could make it easier to model harmonic movement via angular coordinates. In short, my major concern is that enforcing too many constraints inspired by the physics rules we already know introduces a bias, which in return affects the generalization of the model.

My final remark is about the EM algorithm in the paper. I think that the details about the EM should be written more clearly and thoroughly. I recommend writing the generative model explicitly which could improve the clarity of the paper in general, as well as allowing different inference algorithms to be applicable.

---

> ### Author Response · Authors · 2020-11-16
> **We thank R4 for the helpful feedback, and appreciate the positive comments on the writing and contribution.**
>
> We also understand the challenges R4 raises, and we address them in turn below. Before addressing them in detail, however, it is important for us to clarify that some of these challenges rest on a misidentification of "physics" (as a very general field of study) with "intuitive physics" (a specific cognitive ability, referring to the commonsense way that people reason about the dynamics of everyday objects and scenes; see [1] for example). Our focus in this paper is on intuitive physics, which is more restricted than physics in general but remains a key common-sense ability outside the grasp of current machine learning systems.
>
> Re. *"However, this can be also seen as a compromise from the generality of the model, since such a predefined grammar might be too restrictive and it might necessitate adding new rules for each new physical phenomena. For instance, in the fields related to optics or thermodynamics, to be able to model phenomena such as refraction-diffraction of light or heat transfer, one might need to modify the grammar manually."*
>
> We respectfully disagree with R4’s claim that our grammar is too restrictive. Our grammar is very general and is designed to handle a wide range of **intuitive physics** scenarios such as the six cases surveyed in [2]. The examples that R4 is providing are from general physics (that may have been used in the scientific discovery tasks) and NOT intuitive physics scenarios.
>
> Re. *"Another drawback of the methodology could be enforcing position and velocity vectors to reside in the Cartesian coordinate system. Such a choice limits the expressiveness of the model, for instance allowing a general coordinate system could make it easier to model harmonic movement via angular coordinates."*
>
> As mentioned above, BSP is a method for intuitive physics reasoning and does not target general physical learning. With respect to the specific example, in fact, it has been shown (examples B,C and D in [1]) that intuitive physics reasoning often leads to the wrong prediction in scenarios involving angular motion. This is why we chose the Cartesian coordinate, since for the tasks of intuitive physics it is the most general system.
>
> Re. *"My final remark is about the EM algorithm in the paper. I think that the details about the EM should be written more clearly and thoroughly. I recommend writing the generative model explicitly which could improve the clarity of the paper in general, as well as allowing different inference algorithms to be applicable."*
>
> Thanks for this suggestion. We have added a complete generative process in Appendix A. We also mention the possible applicability of other inference methods in the revisited draft.
>
> Re. *"Nevertheless, experimental sections can be extended by large-scale applications or real world datasets. In other words, learning the underlying physical rules from a data set which is collected through real-world sensors would provide further evidence to the impact of such a framework."*
>
> Our experiments are in line with the current state-of-the-art methods and existing literature in intuitive physics reasoning [3,4,5]. This allows us to fairly compare our method with them. But we agree that scaling models of intuitive physics to large scale datasets will be very useful and important. Unfortunately, we are not aware of such publically available datasets and it is beyond the scope of this paper to construct one. If the reviewer is aware of any such dataset, we are more than happy to try it for the camera ready version, if accepted.
>
> Re. *"Although relation to prior work is clearly addressed in the paper, authors compare their results to only OGN."*
>
> To the best of our knowledge, OGN is the most competitive neural model to ours which subsumes other neural baselines such as interaction networks (INs) and builds on top of them. If the reviewer suggests adding INs is still helpful, we are happy to add it.
>
> ---
>
> [1] Battaglia, Peter W., Jessica B. Hamrick, and Joshua B. Tenenbaum. "Simulation as an engine of physical scene understanding." Proceedings of the National Academy of Sciences 110.45 (2013): 18327-18332.
>
> [2] Kubricht, James R., Keith J. Holyoak, and Hongjing Lu. "Intuitive physics: Current research and controversies." Trends in cognitive sciences 21.10 (2017): 749-759.
>
> [3] Sanchez-Gonzalez, Alvaro, et al. "Hamiltonian graph networks with ode integrators." arXiv preprint arXiv:1909.12790 (2019).
>
> [4] Battaglia, Peter, et al. "Interaction networks for learning about objects, relations and physics." Advances in neural information processing systems. 2016.
>
> [5] Ullman, Tomer D., et al. "Learning physical parameters from dynamic scenes." Cognitive psychology 104 (2018): 57-82.

---

### Author Response · Authors · 2020-11-16
**Common Response**

We thank all reviewers for their feedback and for recognizing our core contributions. In addition to individual replies that directly address their individual concerns and questions, we also would like to summarize the main changes in the revised draft: We have added an additional paragraph to discuss related work in symbolic regression, and two new experiments, which were requested by multiple reviewers:

1. The first experiment added to the new Section 4.2 (titled "Generalization") is to show the generalisation ability of our proposed method beyond the training data. Our method generalises to completely different scenarios as expected, especially when true laws are discovered, but also when **approximate** laws are discovered. The new experiment shows how an **approximate law** from our model can be easily transferred to a different environment than the one it was trained on. In brief, we show that an approximate law learnt from the BOUNCE scenario where the gravity was pointing inwards to the screen can be successfully transferred without any additional "training" to a setup where the gravity is pointing in downward direction. Please see more details in Section 4.2 of the revised draft.
2. Secondly, we added a new experiment to the section "Learning with Unobserved Properties" (previously Section 4.2 and now Section 4.3). We show how the EM algorithm can be applied to carry out joint learning and inference in the MAT scenario. This additional experiment, together with the existing one, gives evidence that the complete EM can work in a diverse set of settings.

We hope these additional results resolve the concerns about the generalisation of our method and make the experiments more diverse.

---

### Comment · ~Ernest_Davis1 · 2021-11-10
**Significant limitations in the  physical model**

The physical model in the paper has significant limitations and some mention of that should be made in the section on "Limitation".

The paper is motivated, in its opening sentences, with the following example "Imagine a ball rolling down a ramp. If asked to predict the trajectory of the ball, most of us will find it fairly easy to make a reasonable prediction. Not only that, simply by observing a single trajectory people can make reasonable guesses about the material and weight of the ball and the ramp."

First, this isn't right. The trajectory is independent of the mass and material of the ball, and it is certainly independent of the mass and material of the ramp (ignoring air resistance and elastic deformation, which are not in the scope of the paper and which are usually unobserved in practice.)

More seriously, a ball rolling on a ramp would seem to be a simple enough scenario, but it is outside the scope of the class of physical theories considered in this paper. The model of motion in section 3.1 does not include rotation, so balls cannot roll, they can only slide. As far as I can tell, forces are assumed to be applied at the center of mass of the objects --- there is certainly nothing in the paper to suggest otherwise --- so the torque that the ramp exerts on the ball that causes it to roll cannot be represented.

In section 3.2, which describes the "grammar of Newtonian physics", the shape of an object is specified by what seems to be a single numerical parameter $s_{i}$. The paper is not specific about it, but I would guess that the shapes of all moving objects are assumed to be circular/spherical and this is the radius. So the ramp cannot be represented as an object, merely as a constraint. The only moving objects allowed in the model are balls.

Can this model be extended to include objects of arbitrary shape (within reason), rotational motions, and forces exerted at a point or across a region of contact?

Finally the NBODY example is not an ecologically valid cognitive task. The change in trajectory due to the gravitational force between two ordinary-sized objects is much too tiny to be perceptible  --- measuring gravitational force between ordinary sized objects requires an ingenious and indirect experimental set up. Inferring the form of the gravitational force from terrestrial observations of the heavenly bodies required the combined genius of Kepler and Newton as well as decades of extremely careful measurements; it's not a thing that experimental subjects are going to be able to do or that would have bestowed any evolutionary advantage.

I note that, though this was rejected by ICLR-2021, it was accepted at NeurIPS 2021.

---

### Decision · Program_Chairs · 2021-01-07
**Final Decision**

**Decision:**

Reject

**Comment:**

This paper proposes a method for learning physics combining symbolic computation and learning in an interesting way, targeting sample efficiency. At the initial evaluation, it was on the fence but leaning towards acceptance, with 3 slightly positive and one slightly negative review.

The strengths lie in the combination between symbolic reasoning and statistical ML with a formulation around the classical EM framework. On the other hand, an important issue of the paper is its quite simplistic evaluation on now very easy problems and benchmarks. While benchmarks tend to be simple in the field of learning physics, current work does address more difficult problems than the problems tackled in this paper.

Another issue discussed was the simple trade-off in injecting hand-crafted inductive bias into a system leading to increased sample efficiency, which was perceived as unsurprising by some reviewers. While this is common in ML, and even strongly more so in learning physics from data synthetically generated with known physical laws, it was perceived to be particularly unsurprising in this paper where the benchmarks are indeed very simple and the laws directly encoded.

The AC discussed this paper with the PCs, and it was judged that the weaknesses in evaluation, in particular the simplicity of the tasks, cannot compensate for the interesting hybrid symbolic/ML formulation, and decided to reject the paper.